# The Effects of Natural Epigenetic Therapies in 3D Ovarian Cancer and Patient-Derived Tumor Explants: New Avenues in Regulating the Cancer Secretome

**DOI:** 10.3390/biom13071066

**Published:** 2023-07-01

**Authors:** Rebeca Kelly, Diego Aviles, Catriona Krisulevicz, Krystal Hunter, Lauren Krill, David Warshal, Olga Ostrovsky

**Affiliations:** 1Department of Gynecologic Oncology, MD Anderson Cancer Center at Cooper University Hospital, Camden, NJ 08103, USA; kelly-rebeca@cooperhealth.edu (R.K.);; 2Cooper Medical School of Rowan University, Camden, NJ 08103, USA; 3Cooper Research Institute, Cooper University Healthcare, Camden, NJ 08103, USA

**Keywords:** natural epigenetic compounds, ovarian cancer, secretome, tumor microenvironment, organoids, spheroids

## Abstract

High mortality rates in ovarian cancer have been linked to recurrence, metastasis, and chemoresistant disease, which are known to involve not only genetic changes but also epigenetic aberrations. In ovarian cancer, adipose-derived stem cells from the omentum (O-ASCs) play a crucial role in supporting the tumor and its tumorigenic microenvironment, further propagating epigenetic abnormalities and dissemination of the disease. Epigallocatechin gallate (EGCG), a DNA methyltransferase inhibitor derived from green tea, and Indole-3-carbinol (I3C), a histone deacetylase inhibitor from cruciferous vegetables, carry promising effects in reprograming aberrant epigenetic modifications in cancer. Therefore, we demonstrate the action of these diet-derived compounds in suppressing the growth of 3D ovarian cancer spheroids or organoids as well as post-treatment cancer recovery through proliferation, migration, invasion, and colony formation assays when compared to the synthetic epigenetic compound Panobinostat with or without standard chemotherapy. Finally, given the regulatory role of the secretome in growth, metastasis, chemoresistance, and relapse of disease, we demonstrate that natural epigenetic compounds can regulate the secretion of protumorigenic growth factors, cytokines, extracellular matrix components, and immunoregulatory markers in human ovarian cancer specimens. While further studies are needed, our results suggest that these treatments could be considered in the future as adjuncts to standard chemotherapy, improving efficiency and patient outcomes.

## 1. Introduction

Despite countless advancements in care and treatment, ovarian cancer continues to be a challenging and deadly disease. In the United States alone, 19,880 new cases of ovarian cancer were estimated to be diagnosed in 2022. Overall, ovarian cancer ranks fifth among causes of death in women, leading to more deaths than any other gynecologic cancer [1,2,3]. The current standard treatment comprises a combination of cytoreductive surgery, with or without platinum-based chemotherapy [3,4,5]. While initial management aims to provide a cure, diagnosis at advanced stages, alongside resistance and recurrence of disease, have posed challenges to the optimal management of ovarian cancer [6,7]. Therefore, there is a pressing need for new and effective treatment avenues in the field of oncology, targeting modalities such as epigenetics and immunotherapies.

Given that mortality rates in ovarian cancer have been linked primarily to chemoresistance and metastasis of occult microdisease [8], epigenetic therapies offer a promising solution to this treatment predicament [9,10,11]. Recent studies have shown that both cancer propagation and chemoresistance arise not only from genetic mutations but also from aberrant epigenetic changes known to involve mechanisms of DNA methylation and histone deacetylation [12,13,14,15]. While cancer cells carry the ability to secrete growth factors that induce additional carcinogenic growth, they also reprogram the heterogenous omentum microenvironment, inducing aberrant epigenetic changes in previously normal surrounding cells [16,17,18,19]. Within this heterogeneous microenvironment, omentum adipose-derived stem cells (O-ASCs) play a crucial role in further inducing chemoresistance and recurrence of disease by secreting protumorigenic cytokines and growth factors [19,20,21,22,23]. Therefore, as previously demonstrated in the literature, epigenetic therapies can not only repair but also reprogram epigenetic aberrations involved in chemoresistant or recurrent disease [24,25,26,27,28,29], leading to new solutions to ongoing treatment challenges.

Regarding the effectiveness of synthetic epigenetic compounds in the treatment of ovarian cancer, in vitro results have been promising; however, their clinical application has been challenged by unexpectedly high toxicity, indicating the need for additional epigenetic treatment options [30,31,32]. Most recently, studies with a pan-histone deacetylase epigenetic compound known as Panobinostat (Pano) have shown favorable results [33,34]. Panobinostat is effective in inhibiting ovarian cancer cells both in vitro and in xenograft ovarian models while significantly decreasing the viability of both Taxol-sensitive and resistant ovarian cancer cells [33,34]. Subsequent studies to determine the benefits of Panobinostat, when used alone or in combination with other regimens, are still needed.

While synthetic epigenetic therapies may have great potential for helping overcome ovarian cancer chemoresistance and recurrence, natural sources of epigenetic drugs have recently been of heightened interest [35,36,37,38,39,40]. Extracted from sources present in our daily diet, they have demonstrated antitumor properties, carrying the prospect of lower toxicity levels and easier patient availability. Epigallocatechin gallate (EGCG), a polyphenol extracted from green tea, is a potent antioxidant and DNA methyltransferase inhibitor (DNMTi) with antineoplastic properties in multiple types of cancer [41,42,43,44]. Regarding its anti-neoplastic properties, EGCG has been associated with the inhibition of cancer growth, migration, invasion, and angiogenesis [45], effects likely related to its ability to reverse, amongst other things, aberrant epigenetic modifications in tumors and their tumorigenic microenvironments. In a similar manner, indole-3-carbinol (I3C), a natural histone deacetylase inhibitor (HDACi) found in cruciferous vegetables such as broccoli, has also shown antitumorigenic properties [37,40,46,47]. I3C improved progression-free and overall survival in patients with high-grade serous ovarian cancer when combined with standard platinum-based chemotherapy [35,48]. Moreover, it has shown promising evidence as a cancer preventive option for breast and prostate cancer [40,49].

Therefore, considering the ease of availability of natural epigenetic components when compared to synthetic compounds, which can be added to a patient’s diet and potentially improve clinical outcomes in recurrent and resistant ovarian cancer, we aimed to investigate the efficacy of natural vs. synthetic epigenetic compounds, with or without chemotherapy, on 3D spheroid and organoid models. Further, we additionally tested the influence of our epigenetic treatments in a more clinically relevant model, or human ovarian cancer specimen, in order to investigate the effects of our treatments on the secretion of growth factors, cytokines, and components of the extracellular matrix, crucial to tumor progression in the presence of a tumorigenic microenvironment in patients.

## 2. Materials and Methods

### 2.1. Therapeutic Agents

Paclitaxel and cisplatin (CT, standard chemotherapy), epigallocatechin gallate (EGCG, a polyphenol derived from green tea), indole-3-carbinol (I3C, a compound derived from cruciferous vegetables), and Panobinostat (Pano, a synthetic epigenetic compound) were used in this study. They were supplied in powdered form and stored at −20 °C. Stock solutions of these compounds were prepared using 100% dimethyl sulfoxide (DMSO; Sigma-Aldrich), while the Cisplatin solution was prepared using PBS. Paclitaxel was supplied by Sigma-Aldrich (St. Louis, MO, USA), while Cisplatin, Panobinostat, EGCG, and I3C were supplied by Cayman Chemical (Ann Arbor, MI, USA). Initial drug concentrations were taken from published concentrations available in the literature [22,29,46,50], with subsequent experiments being performed to determine dose dependence on 3D ovarian tumor growth. The dose noted to be most efficacious in downregulating cancer growth was subsequently used in the remainder of our experiments.

### 2.2. Tissue Culture

Caov-3, a platinum-sensitive recurrent human ovarian cancer cell line (American Type Culture Collection, Manassas, VA, USA), was grown in 3D culture. The cells were cultivated in Dulbecco’s modified Eagle’s medium (DMEM; ThermoFisher Scientific, Waltham, MA, USA) with 10% FoundationTM fetal bovine serum (FBS; GeminiBio, West Sacramento, CA, USA) and 1% Gibco^®^ antibiotic–antimycotic (ThermoFisher Scientific). Media changes occurred every 2–4 days, with cell splitting being performed when 80–90% cell confluence was reached. Additionally, 70% Caov-3 cells mixed with 30% omentum-derived tumorigenic stem cells [22] extracted from a patient’s omentum (IRB # 20-020EX) with a diagnosis of stage III serous ovarian cancer were simultaneously grown and treated in 3D culture, named organoids. This allowed for the investigation of the effects of heterogenicity in cancer cells and of a tumorigenic environment in epigenetic and standard chemotherapy treatments. All cells were incubated at 37 °C.

#### 2.2.1. Establishing 3D Tissue Cultures

Caov-3 cells were grown in culture flasks as described above. Subsequently, the hanging-drop method was used to produce 3D spheroids at a density of 1600 cells per drop [29,51]. Spheroids were individually transferred and plated in 9 or 24 bacterial well plates. On days 0 and 3 following transfer, triplicates or tetraplicates were exposed to treatments of EGCG, I3C, Pano, and cisplatin/taxol (CT). The control group was grown in the same conditions as the treatment groups, in the absence of chemotherapy or drugs but in the presence of DMSO or PBS. Spheroids’ initial diameter was approximately 300–400 µM as previously published in our lab [29]. During the first experiment, the effect of two different doses of EGCG (10 and 50 µM) and I3C (50 and 200 µM) on spheroid growth was investigated following a literature analysis of IC50 or optimal tumor suppression doses for these compounds [22,29,46,50]. The hanging drop method described above was additionally utilized to produce 3D structures of 70% Caov-3 cells mixed with 30% omentum derived tumorigenic stem cells, labeled as organoids [51,52,53]. Subsequently, both spheroids and organoids were exposed to epigenetic monotherapies as well as combination treatments of CT (Cisplatin 7.5 µM and Paclitaxel 5.4 nM) with our epigenetic treatments. Two completely independent experiments were performed, both with spheroid and organoid models, with triplicates or tetraplicates in each experiment.

#### 2.2.2. Measurement of Spheroid and Organoid Growth in 3D Culture

(A) Calculations of 3D growth: Throughout the experiments, three-dimensional spheroid and organoid images were captured on days 0, 5, 7, 10, 14, and 17 with an inverted microscope (Leica Microsystems), allowing for evaluation of morphology and growth. Image J was utilized to obtain diameter measures, from which the average radius of each treatment group was derived, as previously reported in the literature [29]. Next, the volume of spheroids and organoids was calculated using the formula *V =* 4/3 *πr*^3^. The percent growth of each 3D structure at various time points in the experiments was then obtained in relation to the initial volume on day 0, according to the following formula [29]:Percent Growth = (current spheroid volume − spheroid volume on day 0)/(spheroid volume on day 0) × 100.

(B) Shrinkage or negative downregulation of growth: As we began to treat spheroids and organoids, image capture was obtained on day 0, followed by the calculation of percent growth on the subsequent experimental days listed above. On day 0, spheroids and organoids all had an initial diameter of approximately 300–400 µM. In some experiments, the efficiency of our drugs led to a shrinkage or reversal of growth in volume following treatment, with subsequent 3D volumes or diameters smaller than those initially recorded on day 0. In these experimental outcomes, we identified this phenomenon as a shrinkage, reversal, or downregulation of growth.

### 2.3. Cellular Assays

#### 2.3.1. Cell Viability, Apoptosis, and Necrosis

Following up to 17 days of various treatments, the remainder of the pre-treated spheroid and organoids underwent viability, apoptosis, and necrosis assays. The cell viability of 3D cancer was measured by the Invitrogen™ Live/Dead viability/cytotoxicity for mammalian cells kit according to the manufacturer’s instructions. Apoptosis assays were performed using the Invitrogen™ eBioscience™ Annexin V-FITC Apoptosis Detection Kit (ThermoFisher Scientific), while the Invitrogen™ eBioscience™ Propidium Iodide Staining Solution Kit (ThermoFisher Scientific) was utilized to detect necrosis. Cells were imaged with an inverted microscope (Leica Microsystems) using both fluorescent settings (which identify Annexin or apoptotic cells as green and Propidium Iodide or necrotic cells as red) and bright field settings.

#### 2.3.2. Proliferation and Regrowth Ability of Cells

Cells derived from previously treated spheroids and organoids were trypsinized and counted with the cellular counter NC200 to distinguish the number of cells that survived following 17 days of experiments. Next, 3000 cells per well from each treatment were plated in 96-well plates and exposed to normal growth conditions. Following 72 h, a 3-(4,5-dimethylthiazol-2-yl)-2,5-diphenyltetrazolium bromide (MTT) cell proliferation assay kit was utilized, following the manufacturer’s recommendations (Invitrogen™ Vybrant™ MTT Cell Proliferation Assay Kit, ThermoFisher Scientific). MTT labeling was performed with cell plates being analyzed in the SpectraMax M3 (Molecular Devices, Sunnyvale, CA, USA). The resulting absorbance or optical density (OD) at 450nm was obtained and normalized with subtractions of appropriate background measures. For visualization of the ability of surviving cells to regrow, we additionally obtained images with an inverted microscope 7 days following replating of 3D-derived cells (Leica Microsystems).

#### 2.3.3. Migration and Invasion

Additionally, pretreated 3D spheroids and organoid-derived cells were tested for their ability to migrate and invade for 24–48 h using the CytoSelectTM 24-Well Cell Migration and Invasion Assay according to the manufacturer’s instructions (8 µm, Colorimetric Format), Cat# CBA-100C, Cell Biolabs, San Diego, CA, USA). After microscopic images for migration and invasion were obtained with an inverted microscope (Leica Microsystems), labeled cells from membranes were extracted and tested for optical density (OD) at 560 nm at a plate reader.

#### 2.3.4. Observation of Colony Formation Following Treatment in 3D Culture

5690 cells from pre-treated spheroids and organoids were transferred to 100 mm plates (100 cells/mm^2^) and incubated for 14 days in cell-appropriate media as previously published in our lab [29]. After that, plates were washed with PBS and incubated with 0.05% Crystal Violet staining solution (Sigma-Aldrich) for 10 min. Plates were again washed 3 times with PBS and allowed to dry. This process leads to the staining of cellular nuclei with a deep purple color. Once dried, plates were photographed for the presence of cell colonies and analyzed.

### 2.4. Tissue Assays

#### 2.4.1. Cancer Tissue Preparation and Treatments

Portions of the ovarian tumor and omentum were collected from a patient diagnosed with high-grade serous ovarian cancer (IRB # 20-020EX). Consent was obtained from the patient prior to the study according to the ethical standards established by the IRB. No patient identifying information was utilized in this study. Both cancer and omentum specimens were sliced at 300 μm with the vibratome. Slices were subsequently weighed. 500 mg of cancer tissue slices and 500 mg of omentum slices were co-cultured indirectly in each well with M199/DMEM medium with 4% serum with the use of the PELCO PrepEze™ 12-Wellplate Insert (TED PELLA Inc., Redding, CA, USA). Specimens were exposed to the following treatments: control medium, EGCG (50 µM), I3C (200 µM), Panobinostat (10 nM), and cisplatin/taxol. After 72 h, the resulting conditioned media of the specimen was collected, centrifuged, and the supernatant fraction was obtained and stored at −80 °C.

#### 2.4.2. Elisa

Conditioned medium from cancer tissue indirectly co-cultured with omentum and pre-treated with our various treatments was thawed at room temperature. ELISA was performed as per the manufacturer’s instructions (R&D Systems) and tested for the effects of our treatments on the expression of various chemo-resistant and tumorigenic-associated proteins using a specific standard curve for each cytokine or growth factor evaluated.

#### 2.4.3. Protein Array

Conditioned medium from cancer tissue indirectly co-cultured with omentum and pre-treated with our various treatments (see material and methods above) was thawed at room temperature. Protein arrays for human growth factors and cytokines (Abcam ab133997) and components of the extracellular matrix, such as MMPs and TIMPs (Abcam ab134004), were performed as per the manufacturer’s instructions and tested for the effects of our treatments on the regulation of levels of human inflammatory markers, cytokines, and components of the extracellular matrix.

### 2.5. Statistical Analysis

All experiments were repeated at least twice (N = 2), with triplicates or tetraplicates in each independent treatment group. Statistical analysis of dose effects, treatment groups, optical density differences in cellular proliferation (MTT assays), migration, invasion, and expression of markers obtained through ELISA was performed with one-way analysis of variance (ANOVA) testing and post hoc Tukey testing. A *p* = 0.05 was used for statistical significance. Data analysis and graphing were performed using SPSS version 27 (IBM, Armonk, NY, USA) and GraphPad Prism software version 9 for Mac (San Diego, CA, USA).

## 3. Results

### 3.1. Dose-Dependent Effect of Natural Epigenetic Therapies on 3D Ovarian Cancer Growth

First, we focused on investigating the effects of different doses of the natural epigenetic compounds EGCG and I3C on ovarian tumor growth. We initially utilized two different concentrations of EGCG (10 µM, 50 µM) and I3C (50 µM, 200 µM) to investigate optimal effects on ovarian cancer 3D spheroid growth. While higher doses have been used in the literature, the concentrations chosen in our study have been previously demonstrated to represent in vitro significant values shown to exhibit overt anti-tumorigenic properties while maintaining sub-toxic levels [22,29,46,50]. For example, regarding the effects of EGCG, 10 µM has been shown to result in 10–30% inhibition of DNMT activity, while 50 µM results in 80% suppression of DNMT in 2D culture [39]. After review of the effects of I3C, 50 µM has been shown to inhibit breast cancer cells by 65%, while 200 µM inhibits proliferation by 90%, with significant downregulation of target promoter gene (CDK6) activity being noted [46]. Therefore, these concentrations were tested in our 3D model given their significance. Our results indicate that even one treatment of the higher concentrations of these epigenetic compounds has a robust effect on the suppression of 3D ovarian cancer spheroid growth following 7 days of assay when compared to lower concentrations. As observed in Figure 1, when lower concentrations were utilized (EGCG 10 µM and I3C 50 µM), significant 3D cancer growth was noted when compared to spheroids treated with higher concentrations of such compounds. For example, the growth of spheroids treated with EGCG 10 µM was downregulated by only 4%, with no significant differences noted in the control group. In contrast, when spheroids were treated with EGCG 50 µM, growth was significantly downregulated by 90% (*p* < 0.05). Similarly, a low concentration of I3C, or 50 µM, did not suppress spheroid growth relative to the control, while a higher concentration, or I3C 200 µM, significantly inhibited 3D spheroid growth by 91% (*p* < 0.05) relative to the control group. Therefore, higher doses of EGCG and I3C were utilized in subsequent extended spheroid kinetic assays.

Once the dose efficacy of our natural epigenetic compounds was tested, 3D ovarian cancer spheroids were exposed to monotherapy treatments of EGCG 50 µM, I3C 200 µM, Pano 10 nM, and standard chemotherapy (CT) at concentrations previously demonstrated in our lab [22,29] on days 0 and 3, with percent growth obtained in an extended kinetic assay (days 0, 5, 10, 14, and 17, pictured below) and compared to the control (or drug-free medium) group. As demonstrated below, downregulation of ovarian cancer tumor growth is noted both through microscopy (Figure 2) and graphical representation of the data, including statistical analysis (Figure 3).

While the control group demonstrates exponential growth, our natural epigenetic monotherapies ascertain a more efficacious response in suppressing cancer growth when compared both to the control and standard chemotherapy-treated groups. The same was observed in the Pano-treated group. Furthermore, when 3D spheroids were treated with EGCG, I3C, and Pano, an overall reversal of growth, or shrinkage, occurred in relation to the control group in an extended kinetic assay. The same effect, however, was not observed when treatment with standard chemotherapy alone took place. As validated by our statistical analysis, on day 10 (Figure 3, left upper panel) of our experiment, EGCG (96% inhibition of growth, *p* < 0.05), I3C, and Pano (both led to 100% inhibition of growth with further shrinkage or negative downregulation of growth, *p* < 0.05) significantly suppressed 3D ovarian cancer growth as compared to control. On day 14 of the experiment (Figure 3, right upper panel), EGCG, I3C, and Pano more drastically suppressed 3D ovarian cancer growth as compared to control (100% inhibition of growth was noted in all three epigenetic treatment groups, with further shrinkage or negative downregulation of growth noted, *p* < 0.05). Details regarding shrinkage or negative downregulation of growth are stated in “Material and Methods”. The effects of our epigenetic treatments were also noted to be robust relative to the chemotherapy-treated groups both on day 10 (Figure 3, left bottom panel) and day 14 (Figure 3, right bottom panel). While chemotherapy alone has an overall effect in suppressing ovarian cancer growth in relation to the control treated groups (67% on day 10, 33% on day 14), it did not suppress 3D spheroid ovarian cancer growth as efficaciously as compared to our epigenetic treatments (Figure 3, bottom row). As observed in the projection boxes in Figure 3 (top row), no statistically significant difference in efficiency was found between natural and synthetic epigenetic compounds.

### 3.2. Apoptosis and Necrosis Assays in Pre-Treated Spheroids

Given that a reversal of growth, or shrinkage, was noted following treatment with the above epigenetic compounds, we sought to investigate the extent of apoptosis and necrosis resulting from our treatments. This assay additionally allows for direct visualization regarding the retention or loss of overall 3D cancer morphology. As demonstrated by our data in Figure 3, downregulation of growth efficiently took place when ovarian cancer cells were exposed to our various epigenetic monotherapies when compared to CT and control groups. In correlation, EGCG, I3C, and Pano treatments induced programmed cell death (apoptosis/necrosis) and significant loss of overall 3D morphology after 17 days of treatments (Figure 4). Following CT treatments, significant downregulation in 3D morphology is noted when compared to control, with high induction of apoptosis/necrosis being observed. Therefore, our epigenetic compounds efficiently induce programmed cell death in 3D ovarian cancer (Figure 4).

### 3.3. Lateral Spread of Microdisease

In addition to the above results, which indicate that epigenetic therapies efficiently induce downregulation of tumor growth, loss of 3D spheroid morphology, and apoptosis/necrosis of cells, we further examined the role of epigenetic therapies in suppressing the lateral spread of disease. As demonstrated by the appearance of 3D spheroids at day 5 (Figure 5), epigenetic therapies have properties that prevent the lateral dispersion of single cells or tumor particles. The same effect is not visualized in the standard chemotherapy-treated group, with lateral spread of particles or a thin monolayer of disease being noted as early as day 5 after treatments.

This phenomenon is also observed across days of treatments in an extended kinetic assay, with epigenetic therapies progressively inducing symmetric downregulation of tumor growth and lateral spread of disease (Figure 2). As days of treatment progressed, however, a thin monolayer of disease surrounding a central 3D core was noted in spheroids treated with a combination of cisplatin and taxol (CT), suggesting that standard chemotherapy alone may not fully halt microspread of disease in ovarian cancer patients. Moreover, single cells noted to shed from 3D structures in the CT-treated group were noted to lead to de novo formation of spheroids, perhaps demonstrating how microdisease can lead to recurrence and spread of ovarian cancer.

### 3.4. Can Epigenetic Therapies Affect Standard Chemotherapy Efficiency in the Treatment of 3D Ovarian Cancer?

As demonstrated by our prior results, EGCG, I3C, and Pano monotherapies efficiently downregulate 3D spheroid ovarian cancer growth while suppressing lateral dispersion of single cells or tumor particles when compared to standard chemotherapy alone (CT) (Figure 2 and Figure 5). Considering such findings, we further tested the efficiency of our epigenetic treatments when combined with CT in order to assess whether combination treatments can augment or optimize the efficiency of standard chemotherapy in the treatment of 3D ovarian cancer (Figure 6).

When EGCG, I3C, and Pano are combined with chemotherapy, overall suppression of tumor growth is noted in a more efficacious manner than in the CT-treated group. More specifically, the EGCG/CT and Pano/CT combos promulgate a more robust and symmetric inhibition of tumor growth across days of treatments, indicating a downregulatory effect of these combination treatments on 3D spheroid growth. Regarding overall morphology, all three combinations suppress dispersion of the lateral monolayer of tumor particles in relation to the chemotherapy-treated group, as observed above. However, despite the efficiency of Pano/CT in downregulating 3D spheroid growth, shedding of single cells can be noted (Figure 6) to a greater extent in the EGCG/CT and I3C/CT treatment groups. Additionally, Pano/CT appears to promote loss of the overall 3D spheroid morphology when compared to EGCG/CT and I3C/CT treatments (Figure 6, day 17 of treatment). Overall, when our epigenetic treatments are combined with standard chemotherapy, a more robust downregulation of growth is noted when compared to monotherapy treatments.

### 3.5. Does the Tumorigenic Microenvironment Affect the Efficiency of Epigenetic Therapies?

In a clinical scenario, the tumorigenic microenvironment carries the ability to impair the efficiency of standard chemotherapy while promoting cancer growth, chemoresistance, and metastasis of disease. This effect is highly driven by omentum stem cell populations, as previously shown in the literature [18,19,20,21]. Given that stem cells create a supportive environment for cancer, we further investigated the effects of our treatments in 3D ovarian cancer organoids, comprising both ovarian cancer tumor cells and omentum derived stem cells (O-ASCs). Therefore, after combining these two cell populations (see details in material and methods), organoids were yielded through the hanging drop method and initially exposed to monotherapy treatments of EGCG, I3C, Pano, and CT on days 0 and 3 of experiments. Percent growth was obtained in an extended kinetic assay (days 0, 5, 7, 10, 14, and 17) and compared to the control (or drug-free medium) group, following the same model previously applied to the spheroid population. As observed in Figure 7 (days 5, 10, and 17 pictured), although organoids contain two different cell populations, their 3D morphology is similar in appearance and structure to that of spheroids. On day 5 following treatments, our epigenetic treatments and standard chemotherapy (CT) are noted to suppress growth and the lateral spread of disease. However, as experimental days progressed, lateral spread of disease and suboptimal downregulation of percent growth were progressively noted in the CT and control groups on days 10 and 17 of experiments when compared to the monotherapy epigenetic groups. In the Pano group, loss of overall 3D morphology also took place, as noted on day 17 of treatment (Figure 7).

From the perspective of an extended kinetic assay, when organoids were treated with our epigenetic compounds, significant downregulation of growth was induced relative to the control group on day 10 (Figure 8, left panel) by EGCG (98% inhibition of growth, *p* < 0.01), I3C (97% inhibition of growth, *p* < 0.01) and Pano (100% inhibition of growth with further shrinkage or negative downregulation of growth noted, *p* < 0.01). Similar effects were also observed on day 17 of experiments (Figure 8, right panel) following treatment with EGCG (98.2% inhibition of growth, *p* < 0.01), I3C (98.1% inhibition of growth, *p* < 0.01) and Pano (100% inhibition of growth with further shrinkage or negative downregulation of growth noted, *p* < 0.01). While it has been established in the literature that chemotherapy alone efficiently suppresses 3D organoid ovarian cancer growth, 75% inhibition of growth was noted in the CT group on day 10 (Figure 8, left bottom panel), with only 23.2% inhibition of growth noted on day 17 (Figure 8, right bottom panel) of experiments relative to the control group. All our epigenetic treatments also efficiently downregulated tumor growth in relation to the chemotherapy-treated group in our experiments (Figure 8, bottom row). Additionally, live/dead immunostaining was performed on organoid cultures following 17 days of our various treatments, showing the same trend and supporting the results shown in Figure 8 (80% live cells were noted in the control group, approximately 15% live cells were noted in the EGCG and I3C groups, and 3% live cells were noted in the Pano group, data not shown). Therefore, our results suggest that epigenetic therapies can efficiently suppress tumor growth even in the presence of tumorigenic omentum-derived stem cells, known to be crucial in providing support for tumor propagation and chemoresistance of disease.

Once we established the efficacy of our epigenetic monotherapies in suppressing tumor propagation, we further tested their efficacy when combined with chemotherapy. While both epigenetic monotherapy treatments and combo treatments efficiently downregulated 3D ovarian cancer growth in relation to control (Figure 9, top panel, *p* < 0.05 for all treatment groups), when ovarian cancer 3D organoids were exposed to combo treatments of EGCG/CT, I3C/CT, and Pano/CT, a more robust effect on suppression of tumor growth was noted in relation to monotherapy treatments, suggesting additive effects between chemotherapy and epigenetic regimens (Figure 9, bottom panel). As illustrated in Figure 9 (bottom panel), significant downregulation took place when organoids were treated with EGCG/CT (82%, *p* < 0.05) and I3C/CT (90%, *p* < 0.05) in relation to EGCG and I3C monotherapies. Contrarily, no significant differences were noted in the downregulation of tumor growth between the Pano/CT and Pano treatment groups (*p* > 0.05). Overall, our results indicate that a combination of natural epigenetic treatments and chemotherapy efficiently halts cancer growth in a 3D organoid ovarian cancer model.

### 3.6. Do Epigenetic Therapies Suppress Post-Treatment Regrowth, Metastasis, and Migration of 3D Derived Ovarian Cancer Cells?

#### 3.6.1. Post-Treatment Ability of 3D-Derived Spheroid and Organoid Cells to Regrow and Proliferate

So far, our epigenetic treatments have efficiently suppressed 3D ovarian cancer growth both in a spheroid and organoid model. Considering that ovarian cancer patients treated with chemotherapy often experience recurrence of disease, we sought to investigate whether epigenetic treatments affect the ability of pre-treated 3D-derived cancer cells from spheroids and organoids to regrow, metastasize, invade, and produce colony forming units (CFU). In order to evaluate the post-treatment potential for regrowth and proliferation, an MTT assay was performed on 3D-derived ovarian cancer cells from pre-treated spheroids and organoids. As illustrated in Figure 10 (left panel), pre-treatment of spheroids with EGCG, I3C, and Pano suppressed the proliferation of 3D-derived cancer cells by approximately 45%, 66%, and 89%, respectively (*p* < 0.05) relative to the control or drug-free group. Similar effects were observed in organoid-derived cells (Figure 10, right panel), in which the ability of cells to proliferate after our epigenetic treatments was significantly suppressed by approximately 59%, 70%, and 76% in the EGCG, I3C, and Pano groups, respectively (*p* < 0.05). In contrast, pre-treatment with standard chemotherapy did not significantly affect the ability of cancer cells to regrow following treatment, both in the spheroid and organoid models.

Results obtained through MTT assays were further confirmed by microscopic contrast images. Regrowth of 3D-derived spheroid (Figure 11, top rows) and organoid (Figure 11, bottom rows) cells demonstrates a high regrowth ability in the control or drug-free group. In the CT-pretreated group, regrowth ability is mildly decreased when compared to the control group, both in spheroid and organoid-derived cells. Comparatively, pre-treatment with EGCG and I3C monotherapies moderately suppressed recovery of cells, while pre-treatment with Pano robustly suppressed regrowth following treatment in both models, as correlated with MTT data (Figure 10). In order to correlate with the clinical scenario in which epigenetic therapies may hold the potential to augment the effects of standard chemotherapy, we additionally tested the regrowth ability following treatment with a combination of chemotherapy and epigenetic compounds. Combinations of CT with EGCG, I3C, and Pano robustly inhibited the ability of cancer cells to regrow when compared to epigenetic treatments alone in both models (Figure 11).

#### 3.6.2. Post-Treatment Invasion and Migration Ability of 3D-Derived Spheroid and Organoid Cells Following Epigenetic Treatments

The ability of ovarian cancer cells to metastasize is an ongoing and concerning problem. The tumorigenic microenvironment provides a supportive environment for proliferation, migration, and invasion of cancer cells, preventing optimal treatment and cure. Considering that, cell migration and invasion assays (see material and methods) were performed on cells derived from both models. In Figure 12 (left panel, monotherapy), our results demonstrate that the ability of spheroid-derived ovarian cancer cells to invade was only significantly impaired by EGCG (13%, *p* < 0.05) and Pano treatments (20%, *p* < 0.01). Upon addition of standard chemotherapy to these epigenetic treatments, invasion ability was significantly inhibited across all combo treatments when compared to the standard chemotherapy-treated group (Figure 12, left panel, combo therapy). In the organoid derived model, which represents cancer in the presence of a tumorigenic microenvironment, EGCG (*p* < 0.01), I3C (*p* < 0.01), and Pano (*p* < 0.001) significantly downregulated the post-treatment invasion ability of cancer cells by approximately 57%, 21%, and 73%, respectively (Figure 12, right panel, monotherapy). When epigenetic treatments were combined with chemotherapy, the invasion ability of pre-treated 3D organoid-derived cells was significantly downregulated by EGCG/CT and Pano/CT by 70% (*p* < 0.001) and 41% by I3C/CT relative to the control group (*p* < 0.01) (Figure 12, right panel, combo therapy).

The above results are further supported by representative images from the invasion assay (Figure 13).

Next, we tested the effect of our epigenetic treatments on migration capacity in organoid-derived cells. Given the significance of the organoid model to translational studies, we concentrated our migration assay on this model only. As shown in Figure 14 (left panel), EGCG, I3C, and Pano progressively suppressed the ability of cancer cells to migrate relative to the control group by approximately 48.7%, 65%, and 82.6%, respectively (*p* < 0.001). When CT monotherapy alone was used, migration capacity was suppressed by only 27% relative to the control group (*p* < 0.01), as observed both in the left and right panels. In a similar manner, when epigenetic treatments were combined with chemotherapy (Figure 14, right panel), significant suppression of migration capacity was noted following combo treatments with EGCG/CT, I3C/CT, and Pano/CT relative to the CT monotherapy treated group of approximately 45% (*p* < 0.01), 62% (*p* < 0.01), and 81% (*p* < 0.001), respectively.

#### 3.6.3. Post-Treatment Ability of 3D-Derived Cells to Form Colony Forming Units (CFU)

So far, our experiments have assessed the capacity of 3D ovarian cancer cells to regrow, migrate, and invade following treatment with epigenetic monotherapies and chemotherapy alone, as well as following combination treatments. Next, we tested the ability of pre-treated organoid-derived cells to form CFU. As observed below (Figure 15, top row), pre-treatment with epigenetic monotherapies overall suppresses CFU formation capacity when compared to the control group, with robust effects observed in the I3C and Pano-treated groups. Comparatively, when epigenetic treatments are combined with chemotherapy, a robust effect is noted across combination treatments (Figure 15, bottom row). Efficacious suppression of CFU formation is equally observed in the EGCG/CT, I3C/CT, and Pano/CT combo groups in relation to the control and CT groups.

### 3.7. Can Epigenetic Treatments Alter the Expression of Protumorigenic Markers in Ovarian Cancer Human Specimens?

#### 3.7.1. Effects on the Secretion of Protumorigenic Factors by ELISA

The secretome has been established as the major regulator of tumors and their microenvironments [54,55]. The secretion of cytokines, growth factors, and components of the extracellular matrix promotes proliferation, metastasis, and invasion and is also involved in the process of chemoresistance to standard chemotherapy [56,57]. Therefore, we aimed to further investigate how our natural and synthetic epigenetic therapies regulate the secretion of growth factors and cytokines involved in tumorigenesis progression in human ovarian cancer specimens in the presence of a tumorigenic omental micro-environment. As described in material and methods, cancer tissue and omentum obtained from a patient with ovarian cancer were co-cultured in the presence of our epigenetic therapies for 4 days. Conditioned medium from co-cultured specimens was collected and tested for secretion of the following human cytokines: VEGF, IL-6, IL-10, and TNF-α. As demonstrated in Figure 16, secretion of the angiogenic factor VEGF was significantly downregulated by EGCG and I3C (17% and 26%, respectively, *p* < 0.05), with no effects resulting from treatment with Pano or CT. The secretion of IL-10, associated with the induction of proliferation and metastasis, was efficiently suppressed by EGCG (90.1%, *p* < 0.05), I3C (96.7%, *p* < 0.05), Pano (86.3%, *p* < 0.05), and CT (89.1%, *p* < 0.05). Secretion of TNF-α, a proinflammatory marker, was significantly downregulated by all of our treatments, with an overall less robust effect observed following treatment with I3C (26.4%, *p* < 0.05) when compared to the remaining treatments (EGCG 79%, Pano 88.6%, CT 91.3% downregulation, *p* < 0.05; calculations are relative to control). The secretion of IL-6, involved in the induction of proliferation in ovarian cancer, was significantly downregulated by EGCG and I3C (80% and 32.5%, respectively, *p* < 0.05). Given that our epigenetic treatments seem to regulate the secretory machinery, we expanded our investigation to a larger scale, evaluating their effects on additional cytokines, growth factors, and extracellular matrix components through protein arrays.

#### 3.7.2. Effects on the Secretion of Cytokines, Growth Factors, MMPs, and TIMPs by Protein Array

Subsequently, the conditioned medium of co-culture specimens was collected and tested for secretion of human cytokines and growth factors relevant to cancer propagation and chemoresistance through protein arrays (Figure 17).

Protein array data were analyzed using optical density comparisons between control and our various treatments, with a summary of pertinent optical density changes being demonstrated in Table 1. Our natural epigenetic treatments downregulated the secretion of IL-10, associated with induction of proliferation, migration, and immunosuppression in ovarian cancer [58,59], by more than 70%, an effect comparable to that obtained by treatment with the synthetic compound Pano. In contrast, IL-8 secretion, also associated with ovarian cancer proliferation [60,61], was downregulated by only 10–20%. IL-11, correlated with chemoresistance and metastasis of disease [62], was inhibited by 50%, 92%, and 97.8% following treatments with EGCG, I3C, and Pano, respectively. Regarding effects on immunoresponse markers, our treatments suppressed levels of RANTES, known to alter T-cells and natural killer cells’ function [63], by 96%, 71%, and 76.7% following treatments with EGCG, I3C, and Pano, respectively.

Next, the secretion of components of the extracellular matrix known to be involved in cancer progression and metastasis, such as matrix metalloproteases (MMPs) and their inhibitors, tissue inhibitors of metalloproteases (TIMPs), was analyzed by protein array as indicated in Figure 18.

Protein array data of MMPs and TIMPs were analyzed using optical density comparisons between control and our various treatments, with a summary of pertinent optical density changes being demonstrated in Table 2. The secretion of MMP-8, which has been correlated with ovarian cancer’s overall prognosis [56,64,65], was downregulated by 30%, 16%, and 7% by EGCG, I3C, and Pano, respectively. Secretion of MMP-13, known to stimulate ovarian cancer metastasis [65], was suppressed by 31%, 36%, and 34% following treatments with EGCG, I3C, and Pano. As expected, the secretion of TIMPs was overall upregulated by our various treatments (Table 2).

## 4. Discussion

Despite advancements in the treatment of ovarian cancer, high mortality rates continue to occur in patients diagnosed with the disease [1,2,3,4,54]. Given challenges with the optimal management of recurrent and metastatic disease, robust efforts from the scientific community have taken place in the past decade, targeting new avenues and treatment options for ovarian cancer patients [7,9,12]. It has been widely established that not only genetic changes, but also epigenetic aberrations play a critical role in cancer propagation, metastasis, and chemoresistance [10,11,16]. Therefore, epigenetic therapies offer an innovative therapeutic approach to such treatment challenges, offering new strategies for augmenting the efficiency of standard chemotherapy, counteracting chemoresistance, and leading to improved clinical outcomes [27,28,29]. Currently, a few synthetic epigenetic compounds have been investigated through clinical trials. However, given toxicity concerns, they have yet to be optimized in order to become widely recommended treatments [30,31,32]. In contrast, natural epigenetic compounds such as EGCG and I3C offer the promise of lesser toxicity and easier patient availability, rising as an option that could be used prior to, in combination with, or following standard chemotherapy treatment [35,36,37,38,39,40]. Regarding their action, natural epigenetic compounds have been previously explored in a multitude of 2D tissue culture cancer models, revealing great potential for suppressing the propagation of cancer [41,42,43,44]. Our study is unique in that it focuses on further exploring the actions of EGCG and I3C versus the synthetic compound Panobinostat (Pano), with or without standard chemotherapy, on 3D ovarian cancer models or human specimens. Our results suggest that natural treatments are efficacious in suppressing ovarian cancer tissue growth alone as well as in the presence of a tumorigenic microenvironment. Taking a step forward, we additionally demonstrate the role of our epigenetic treatments in regulating the secretory machinery that promotes tumor propagation, metastasis, and chemoresistance in 3D models and human tissue specimens.

Regarding cancer progression, our results support that natural epigenetic compounds efficiently inhibit 3D ovarian cancer tumor growth in a dose-dependent manner (Figure 1, Figure 2 and Figure 3). Interestingly, the efficiency of action of EGCG and I3C was noted not only in spheroids, which represent an ovarian cancer cell population (Figure 1, Figure 2 and Figure 3), but also in organoids (Figure 7 and Figure 8), in which ovarian cancer cells and tumorigenic omental adipose-derived stem cells are combined, representing the complex interactions between the tumor and its heterogeneous tumorigenic microenvironment (Figure 7 and Figure 8). As the tumorigenic microenvironment supports cancer growth, metastasis of disease, and chemoresistance [18,66,67], our results are promising in evidencing that natural epigenetic therapies can reprogram the tumorigenic microenvironment regarding cancer progression, with comparable effects to those elicited by the synthetic compound Pano (Figure 3 and Figure 8). The advantages of EGCG and I3C when compared to Pano, however, include ease of availability independent of provider and the possibility of incorporation into a patient’s diet as a treatment adjunct or even as a prophylactic regimen. In-depth in vivo studies or Phase 1 trials, however, are still needed in order to evaluate the toxicity and safety levels in patients.

Considering that platinum-based chemotherapy in combination with taxol is the first-line treatment recommendation for ovarian cancer patients, it was critical to investigate the efficiency of natural epigenetic therapies in combination with standard chemotherapy. While prior studies have explored the effects of EGCG and I3C in suppressing additional types of cancers in a 2D tissue culture model [39,43,45,46,47,48,68], there is paucity of data regarding the efficacy of these compounds when combined with chemotherapy. As demonstrated by our results (Figure 6 and Figure 9), combination treatments had higher efficacy in suppressing tumor growth when combined with standard chemotherapy in spheroids (Figure 6) and organoids (Figure 9). Contrarily, while Pano is efficacious in suppressing tumor growth as a monotherapy, no significant differences in tumor growth were noted in our study when Pano was combined with standard chemotherapy (Figure 9). Our results therefore suggest a role for natural epigenetic therapies in augmenting or serving as an adjunct to standard of care treatment in ovarian cancer patients.

Given that there is an ongoing challenge in the treatment of patients with recurrent or metastatic ovarian cancer, with rates of recurrence as high as 70% in the first three years following diagnosis [2,5], it was relevant to explore post-treatment outcomes of epigenetic treatments for 3D ovarian cancer. In a clinical scenario, microdiseases can remain in the human body following debulking surgery or chemotherapy treatment, later leading to recurrence or metastasis [8]. To represent this phenomenon, we additionally showed that natural epigenetic compounds have a lasting effect in suppressing tumor regrowth, migration, invasion, and the capacity to form colonies in 3D-derived ovarian cancer cells previously treated with such compounds. Our study shows that 3D-derived cells from spheroids and organoids previously treated with EGCG and I3C have limited capacity to regrow when compared to the chemotherapy-treated groups (Figure 10). When chemotherapy is combined with EGCG and I3C, the post-treatment regrowth ability of cells is further suppressed (Figure 11). Similar effects are observed in cells treated with the synthetic epigenetic compound Pano (Figure 10 and Figure 11). Additionally, natural epigenetic compounds suppress the post-treatment ability of 3D organoid-derived cells to invade (Figure 12 and Figure 13), migrate (Figure 14), and form colonies (Figure 15). These results infer that our epigenetic treatments further potentiate post-treatment effects in ovarian cancer, perhaps by altering aberrant gene expression noted in cancer and its tumorigenic environment.

The secretion of growth factors, cytokines, and components of the extracellular matrix (known as the secretome) has been shown to play a crucial role in regulating progression, resistance, and metastasis in various cancers [54,55,56,57]. We therefore tested the role of our epigenetic treatments in influencing the secretome of ovarian cancer tissue in the presence of a tumorigenic microenvironment, given that this model most closely represents the clinical progression of disease (Figure 16, Figure 17 and Figure 18, Table 1 and Table 2). Our results indicate that EGCG, I3C, and Pano significantly downregulate levels of the protumorigenic cytokine IL-11, which is a major factor implicated in the chemoresistance of ovarian cancer cells through the activation of the JAK2-STAT5 pathway [62]. Considering that targeting this pathway can reverse chemoresistance properties and re-sensitize cells to platinum treatment, our natural epigenetic compounds may be promising options for improving the clinical outcomes and prognosis of patients with platinum-resistant ovarian cancer by downregulating levels of IL-11 [62]. Similar to the effects observed following treatment with Pano, EGCG and I3C downregulate the secretion of IL-10, an immunosuppressive factor in ovarian cancer ascites correlated with proliferation and metastasis, as well as maintenance of the tumorigenic microenvironment [58,59]. Therefore, inhibition of IL-10 by our epigenetic treatments can target the growth and metastasis of ovarian cancer cells. IL-6, an inflammatory chemokine associated with tumorigenesis and chemoresistance in disease, was also significantly downregulated by EGCG and I3C [61]. Moreover, critical factors involved in the metastasis of disease and overall prognosis (MMPs) and their inhibitors (TIMPS) were also robustly affected by our treatments [56,64,65]. MMP 8, 10, and 14 were downregulated by EGCG, I3C, and Pano, while levels of TIMPs 1 and 3, as expected, were upregulated (Table 1 and Table 2). These results support the idea that natural epigenetic compounds can modulate the secretome of ovarian cancer, serving as new avenues to overcome tumor propagation and metastasis and perhaps aiding in reversing chemoresistance to platinum agents.

Although the focus of this study was not to fully explore the molecular mechanisms of EGCG and I3C, studies available in the literature have demonstrated that suppression of tumor growth can be explained through the induction of apoptosis by upregulation of cell cycle regulatory markers. For example, inhibition of tumorigenic progression by EGCG has been correlated with induction of P53 [45], P16, and P21 [43]. These markers have been widely established as checkpoint inhibitors known to halt cell cycle progression and subsequent tumor growth through apoptosis and cell death. In a similar manner, I3C has been shown to inhibit cancer progression in lung tumors by downregulating levels of cyclin E and its partner CDK2, enhancing apoptotic cell death [47]. Given that EGCG and I3C induced apoptotic cell death in our 3D models (Figure 4), downregulating tumor growth, we hypothesize that the same molecular mechanisms also took place. Further studies uncovering more in-depth molecular mechanisms of EGCG and I3C in ovarian cancer are, however, still needed.

It is also important to highlight that while prior concerns existed regarding hepatocellular toxicity inflicted by EGCG in humans, the pre-FRIEND trial has evaluated the safety of EGCG in premenopausal women, with no drug-induced liver injury demonstrated in any patients [69]. Regarding I3C, most human studies have failed to report overt toxicity [70,71], suggesting that natural epigenetic compounds are plausible treatment options in humans.

## 5. Conclusions

To our knowledge, this is the first study to show the high efficiency of natural epigenetic treatments in suppressing 3D ovarian cancer growth and post-treatment recovery, invasion, and migration of disease in the presence of a tumorigenic microenvironment. Additionally, natural epigenetic compounds can regulate the protumorigenic secretome, which is involved in metastasis, chemoresistance, and relapses of disease. While further studies are needed, the effects obtained from natural epigenetic therapies and their limited toxicity when compared to synthetic epigenetic compounds further support their future potential as prophylactic, adjunct, or therapeutic options for improving the clinical outcomes of ovarian cancer patients.

## Figures and Tables

**Figure 1 biomolecules-13-01066-f001:**
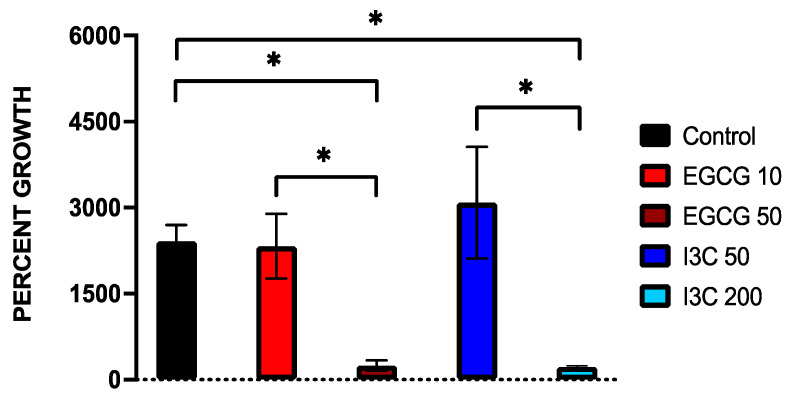
Effects of dose concentration on spheroid growth. Spheroids were treated with different concentrations of EGCG (10 µM, 50 µM) and I3C (50 µM, 200 µM) in a single dosage and allowed to grow for 7 days. Percent growth was obtained as a measure of outcome over time, with robust suppression of growth noted in a dose-dependent manner by EGCG 50 µM (*p* < 0.05) and I3C 200 µM (*p* < 0.05) when compared to lower doses of the same compounds. Significant differences were also noted between EGCG 50 µM (downregulation of growth of 90%, * *p* < 0.05) and I3C 200 µM (downregulation of growth of 91%, *p* < 0.05) relative to control. Two independent experiments were performed, including triplicates for each treatment group. The asterisk (*) represents *p* < 0.05.

**Figure 2 biomolecules-13-01066-f002:**
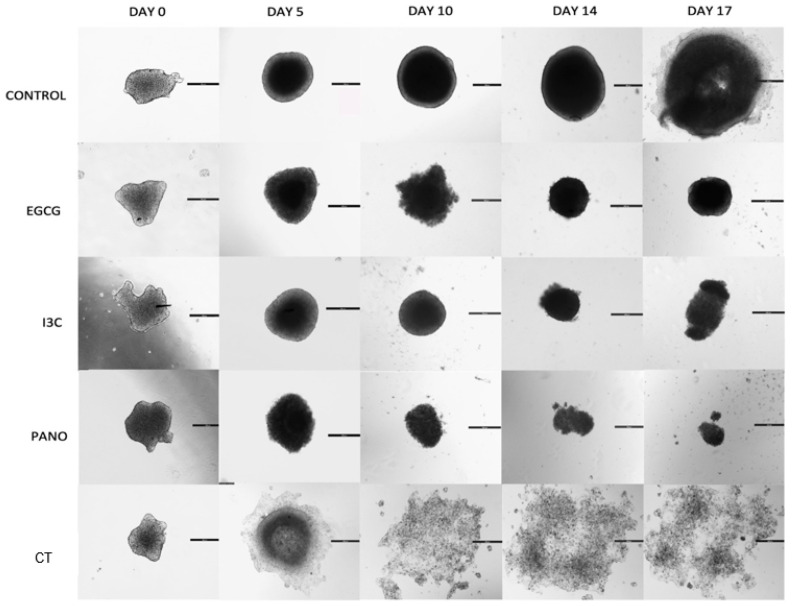
Microscopic representation of 3D ovarian cancer growth after epigenetic monotherapy treatments. The assay includes treatments with EGCG, I3C, and Pano which effectively suppress spheroid 3D growth over time when compared to control conditions and standard chemotherapy (CT). Additionally, these epigenetic compounds are noted to provide overall uniform shrinkage of spheroids in the absence of the spread of peripheral tumor microdisease. Two independent experiments were performed, including triplicates or tetraplicates for each treatment group.

**Figure 3 biomolecules-13-01066-f003:**
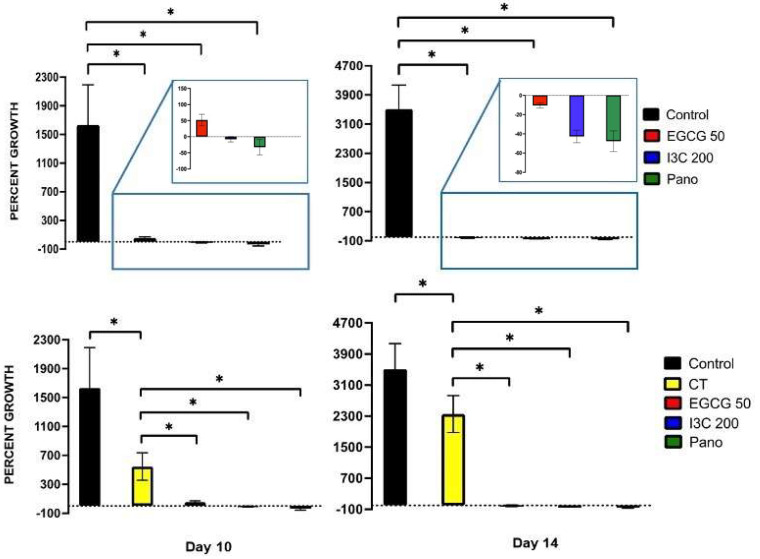
Effects of epigenetic monotherapies and standard chemotherapy (CT) on spheroid growth in a prolonged kinetic assay on days 10 (**left panel**) and 14 (**right panel**). Ovarian cancer tumor growth is efficaciously suppressed by treatments with EGCG, I3C, and Pano in relation to the control (**top row**) treated groups on days 10 (96% by EGCG, 100% with further negative downregulation of growth by I3C and Pano, *p* < 0.05) and day 14 (100% inhibition of growth was noted in all three epigenetic treatment groups with further shrinkage or negative downregulation of growth noted, *p* < 0.05). CT suppressed ovarian cancer growth in relation to the control-treated groups (67% on day 10, 33% on day 14); however, not as efficiently when compared to our epigenetic treatments (**bottom row**). On the **top row**, a projection box allows for visualization of the effects of our epigenetic treatments. No significant differences were noted between natural compounds and Pano on days 10 and 14, suggesting similar efficiency of action between them (*p* > 0.05). Two independent experiments were performed, including triplicates or tetraplicates for each treatment group. The asterisk (*) represents *p* < 0.05.

**Figure 4 biomolecules-13-01066-f004:**
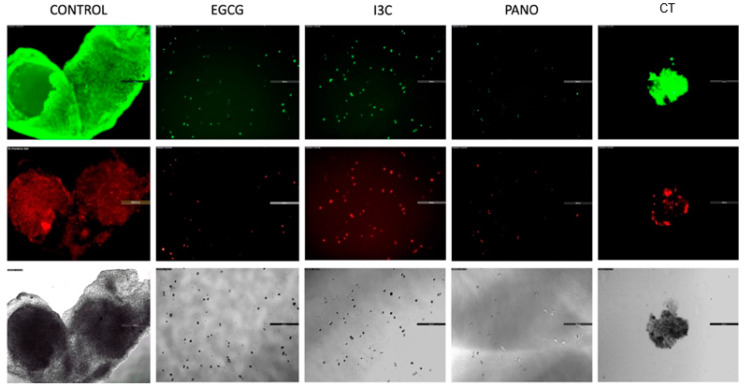
Apoptosis and necrosis assays following 17 days of treatments. Spheroids previously treated with various compounds are processed through an apoptosis/necrosis assay as described in Materials and Methods. The remainder of the spheroid morphology and cell fraction are pictured in green by annexin V (apoptosis, **top panel**) and red by propidium iodide staining (necrosis, **middle panel**). Bright field images (**bottom panel**) are provided for comparison according to treatment groups. Two independent experiments were performed, including triplicates or tetraplicates for each treatment group.

**Figure 5 biomolecules-13-01066-f005:**
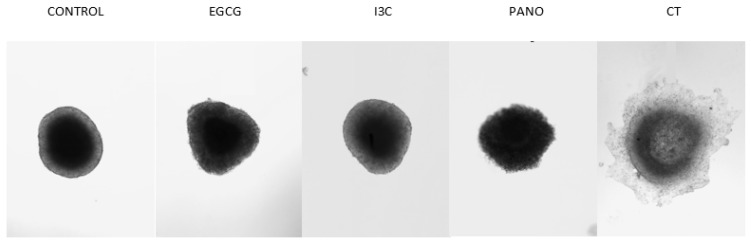
Representation of the lateral spread of microdisease. In 3D culture, both natural (EGCG 50 µM and I3C 200 µM) and synthetic (Pano 10 nM) epigenetic compounds suppress the lateral dispersion of tumor cells or particles. In contrast, the same response is not noted in spheroids exposed to standard chemotherapy (CT), as visualized here on day 5 of experiments. Two independent experiments were performed, including triplicates or tetraplicates for each treatment group.

**Figure 6 biomolecules-13-01066-f006:**
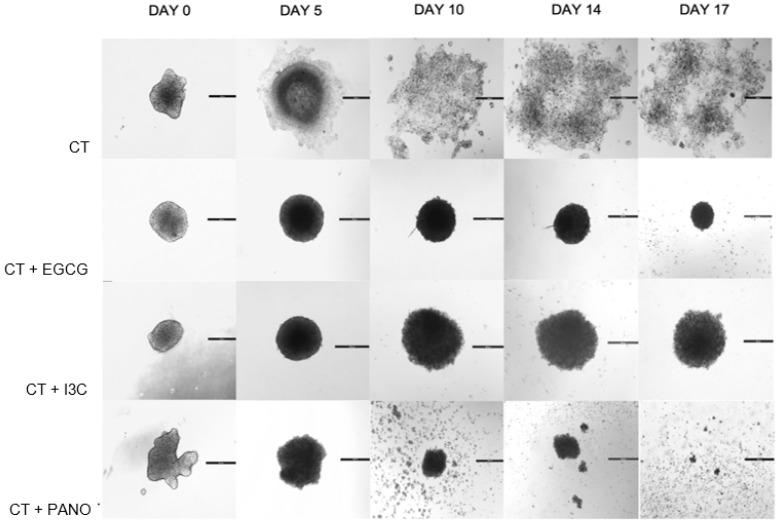
Microscopic representation of 3D ovarian cancer spheroid growth after combo therapy with epigenetic treatments and standard chemotherapy. In 3D culture, combinations of EGCG/CT, I3C/CT, and Pano/CT further suppress tumor growth in comparison to standard chemotherapy (CT) alone. EGCG/CT and Pano/CT have a more robust effect on volume inhibition, while Pano/CT promotes loss of 3D morphology. Lateral dispersion of disease can be visually observed in the CT-treated group. Two independent experiments were performed, including triplicates or tetraplicates for each treatment group.

**Figure 7 biomolecules-13-01066-f007:**
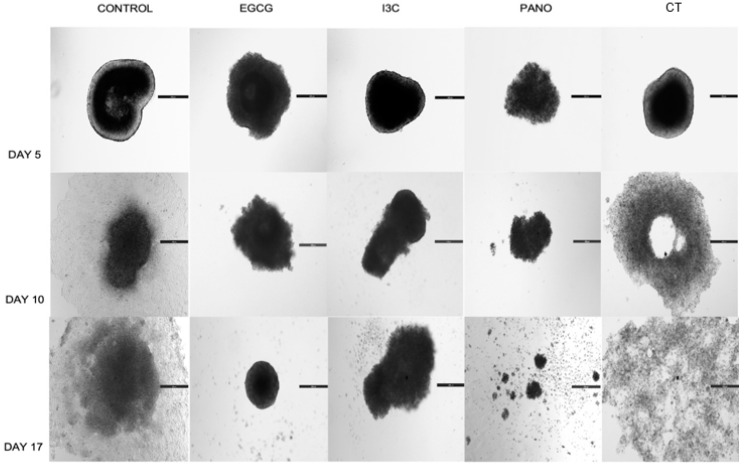
Morphology of organoids in 3D culture following epigenetic or CT treatment. Treatments with natural (EGCG, I3C) and synthetic (Pano) epigenetic compounds were performed on days 0 and 3, with response to therapy recorded in percent growth on days 0, 5, 7, 10, 14, and 17. As illustrated, as the days of the experiment progressed, both the CT and control groups provided suboptimal downregulation of percent growth and lateral spread of disease when compared to the monotherapy epigenetic treatment groups. Two independent experiments were performed, including triplicates or tetraplicates for each treatment group.

**Figure 8 biomolecules-13-01066-f008:**
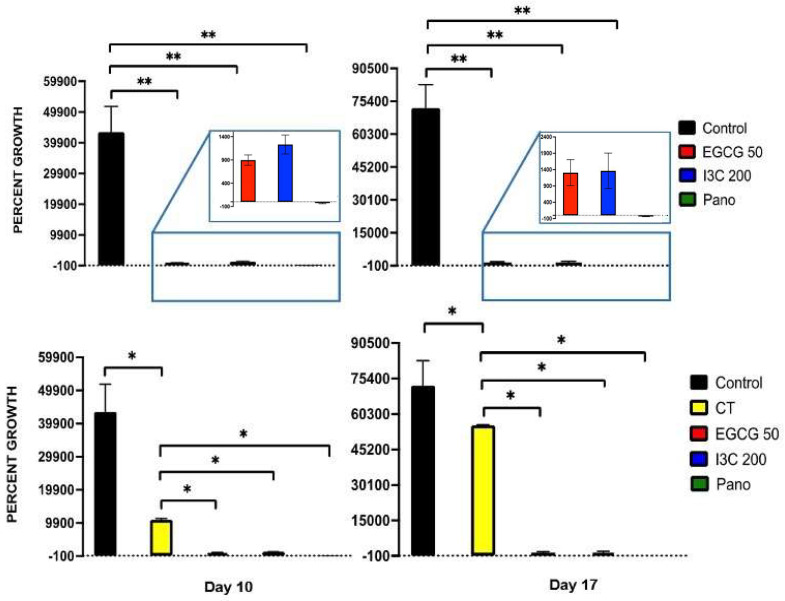
Effects of epigenetic monotherapies and standard chemotherapy (CT) on organoid growth in a prolonged kinetic assay on days 10 (**left panel**) and 17 (**right panel**). Ovarian cancer organoid tumor growth is efficaciously suppressed by treatments with EGCG, I3C, and Pano in relation to the control (**top row**) treated groups on days 10 (98% by EGCG, 97% by I3C, 100% by Pano with further negative downregulation of growth, *p* < 0.01) and day 17 (98.2% by EGCG, 98.1% by I3C, 100% by Pano with further negative downregulation of growth, *p* < 0.01). CT suppressed ovarian cancer growth in relation to the control-treated groups (75% on day 10, 23% on day 14, *p* < 0.05), but not as efficiently when compared to our epigenetic treatments (**bottom row**). On the top row, a projection box allows for visualization of the effects of our epigenetic treatments. No significant differences were noted between natural compounds and Pano on days 10 and 17, suggesting similar efficiency of action between them (*p* > 0.05). Two independent experiments were performed, including triplicates or tetraplicates for each treatment group. The asterisk (*) represents *p* < 0.05 and (**) represents *p* < 0.01.

**Figure 9 biomolecules-13-01066-f009:**
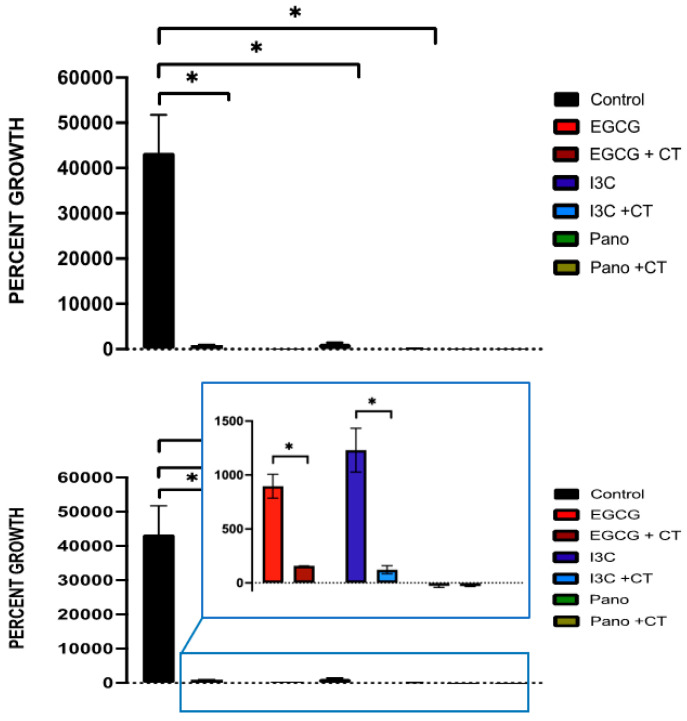
Effects of epigenetic treatments alone or in combination with chemotherapy in 3D ovarian cancer organoids. As indicated in the top panel, organoid’s percent growth was significantly inhibited by treatment with our natural epigenetic treatments when used alone or in combination with chemotherapy relative to the control group (*p* < 0.05 for all treatment groups). Higher efficacy is noted both in the EGCG/CT (82%, *p* < 0.05) and I3C/CT (90%, *p* < 0.05) groups when compared to epigenetic monotherapy treatments on day 10 of experiments. No significant differences were noted when Pano was administered alone or in combination with chemotherapy. Two independent experiments were performed, including triplicates or tetraplicates for each treatment group. The asterisk (*) represents *p* < 0.05.

**Figure 10 biomolecules-13-01066-f010:**
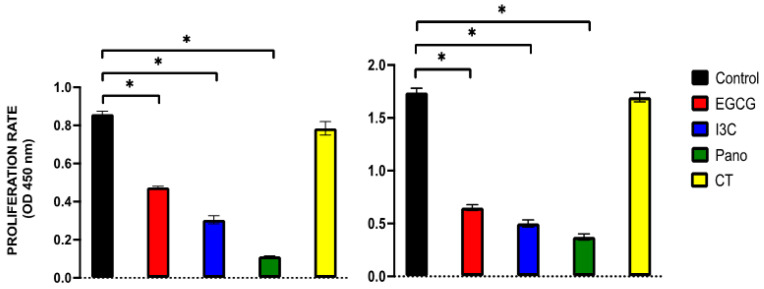
Regrowth ability of cells following epigenetic and chemotherapy treatments in spheroids (**left panel**) and organoids (**right panel**). An MTT assay was utilized to quantify the regrowth ability of 3D-derived cancer cells following treatment with EGCG, I3C, Pano, and CT. Pre-treatment of spheroids and organoids with EGCG, I3C, and Pano significantly suppressed the proliferation of cancer cells following treatments (*p* < 0.05 for all epigenetic groups). Two independent experiments were performed, including triplicates for each treatment group. The asterisk (*) represents *p* < 0.05.

**Figure 11 biomolecules-13-01066-f011:**
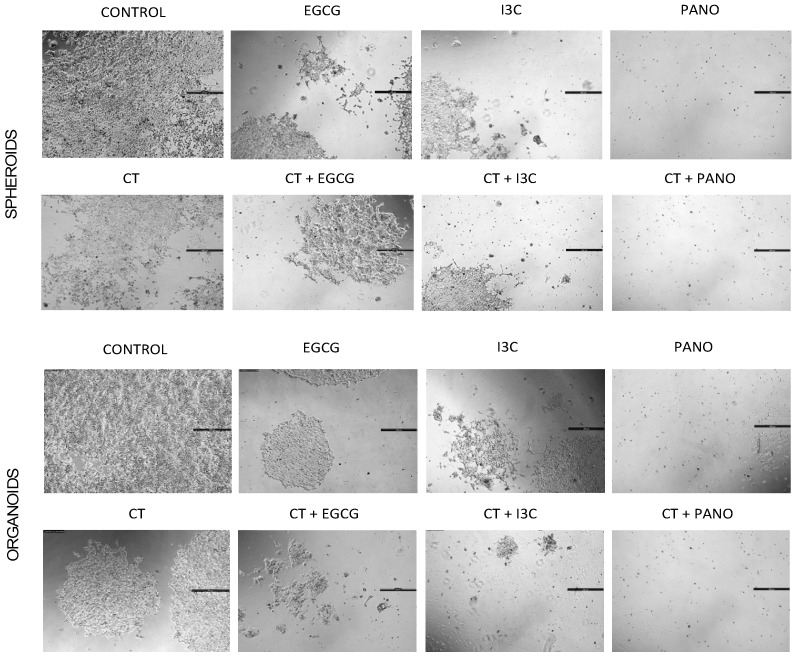
Microscopic representation of the regrowth of a spheroid (**top rows**) derived cells or organoid (**bottom rows**) derived cells of epigenetic versus combo treatments with chemotherapy. 3D-derived cells were plated following 17 days of our various treatments and allowed to regrow in normal conditions for an additional 7 days. Cells in the control group showed high regrowth abilities. In cells pre-treated with epigenetic monotherapies, EGCG and I3C moderately suppressed the recovery of cells in comparison to the control group, while Pano more robustly suppressed growth in both models. The combination of epigenetic treatments with chemotherapy further suppresses regrowth ability when compared to monotherapy treatments. Two independent experiments were performed, including triplicates for each treatment group.

**Figure 12 biomolecules-13-01066-f012:**
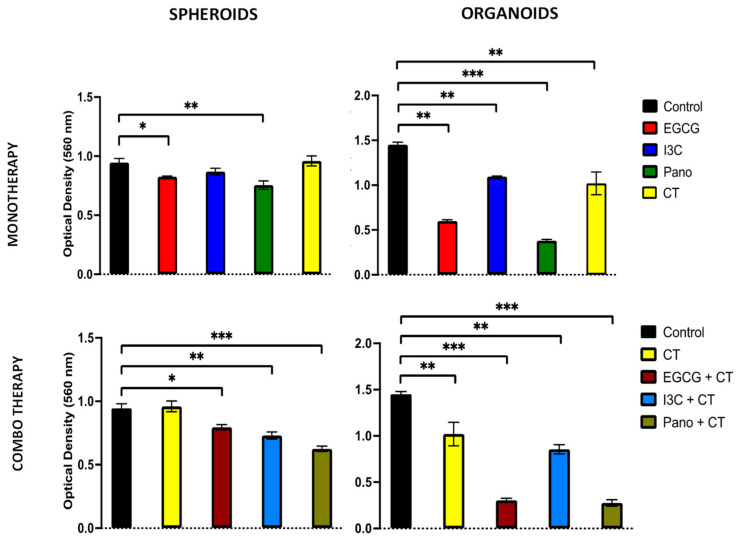
Ability of 3D-derived spheroid (**left panel**) and organoid (**right panel**) cells to invade following epigenetic monotherapy or combo treatments. An invasion assay was used to evaluate the invasion capacity of spheroids and organoids following pre-treatment with various therapies. The ability of 3D-derived organoid cells pre-treated with monotherapy epigenetic treatments was significantly affected by EGCG (57%, *p* < 0.01), I3C (21%, *p* < 0.01), and Pano (73%, *p* < 0.001) relative to control. When combo treatments of EGCG/CT and Pano/CT were used, the post-treatment invasion ability of organoid-derived cells was downregulated relative to standard chemotherapy alone by approximately 70% (*p* < 0.001), with 41% inhibition induced by treatment with I3C/CT (*p* < 0.01). Two independent experiments were performed, including triplicates for each treatment group. The asterisk (*) represents *p* < 0.05, (**) represents *p* < 0.01, and (***) represents *p* < 0.001.

**Figure 13 biomolecules-13-01066-f013:**
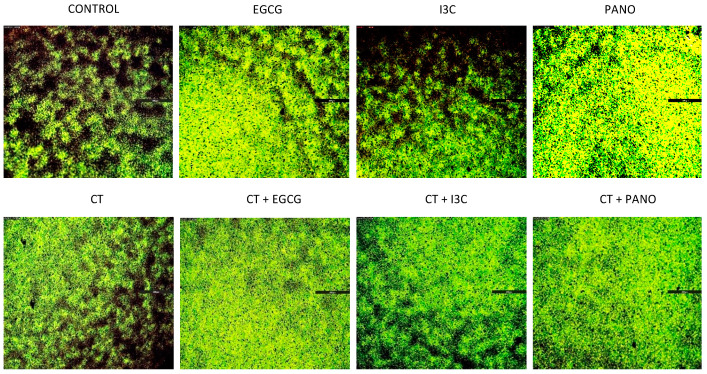
Microscopic representation of invasion assay from post treated 3D derived organoid cells following epigenetic monotherapy (**top row**) or combination (**bottom row**) treatments. In organoids, the ability to invade is significantly affected by pre-treatment with EGCG and Pano, both in monotherapy and groups. The dark color represents cell invasion; see Materials and Methods. Two independent experiments were performed, including triplicates for each treatment group.

**Figure 14 biomolecules-13-01066-f014:**
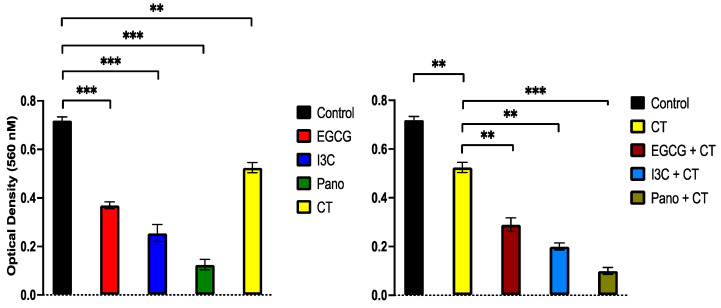
Migration ability of 3D organoid-derived cells following epigenetic monotherapy (**left panel**) or combination treatments with chemotherapy (**right panel**). A migration assay was used to evaluate the capacity of cancer cells to migrate following pretreatment with our various therapies. As observed, epigenetic monotherapies halt the migration of cancer cells. Two independent experiments were performed, including triplicates for each treatment group. The asterisk (**) represents *p* < 0.01 and (***) represents *p* < 0.001.

**Figure 15 biomolecules-13-01066-f015:**
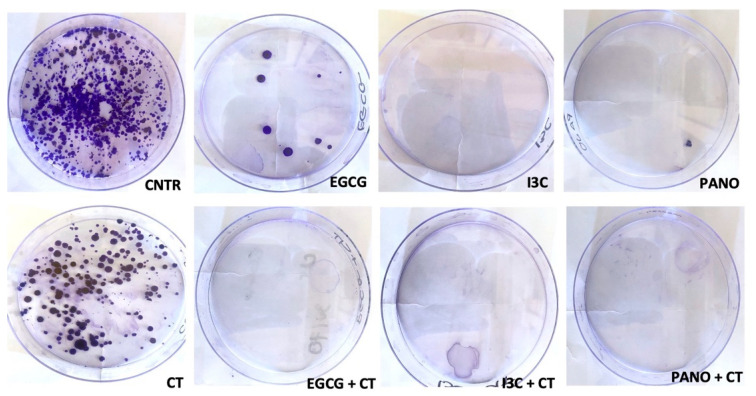
Ability of 3D-derived organoid cells to form CFU following epigenetic monotherapies (**top row**) and combo treatments with CT (**bottom row**). CFU formation assay was performed as described in Materials and Methods. Overall, suppression of the post-treatment capacity of CFU to establish themselves is noted after treatment with epigenetic monotherapies in relation to the control group, as well as in combination treatments in relation to CT alone. Two independent experiments were performed.

**Figure 16 biomolecules-13-01066-f016:**
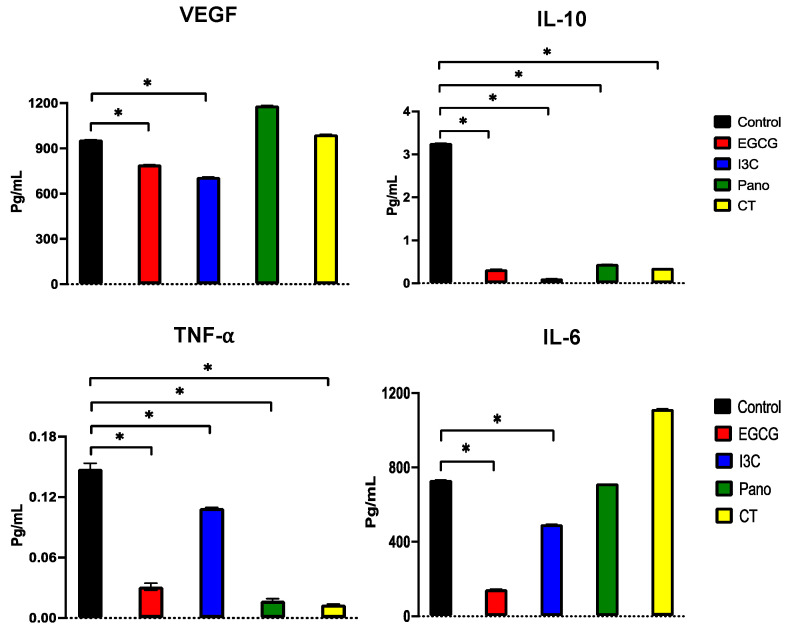
Effects of epigenetic treatments on cytokine secretion in ovarian cancer tissue by ELISA. ELISA assays were performed as described in material and methods. Overall, secretion levels of growth factors and cytokines were determined by a standard curve with appropriate recombinant growth factors. Significant differences are indicated with an asterix (*) in relation to the control or drug-free treatment with *p* < 0.05. Two independent experiments were performed, including triplicates for each treatment group.

**Figure 17 biomolecules-13-01066-f017:**
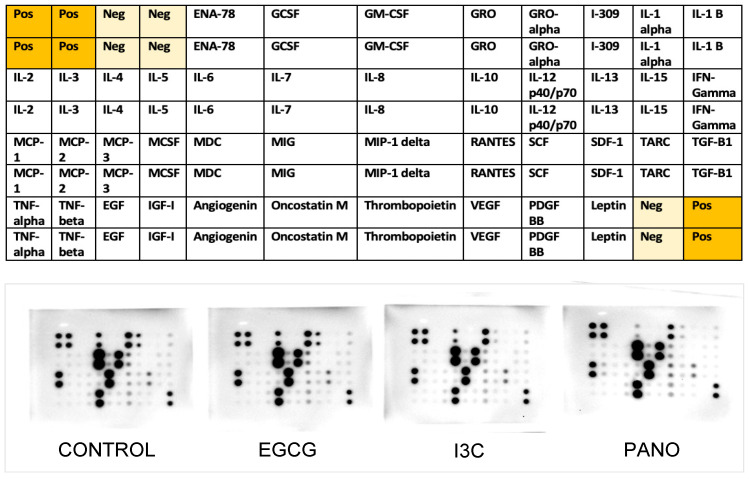
Protein array map of growth factors and cytokines (**top row**) and experimental images (**bottom row**) of human specimens pretreated with epigenetic therapies. Conditioned medium collected from ovarian cancer human specimens pretreated with EGCG, I3C, and Pano was tested by protein array for cytokines and growth factors expression. See Materials and Methods.

**Figure 18 biomolecules-13-01066-f018:**
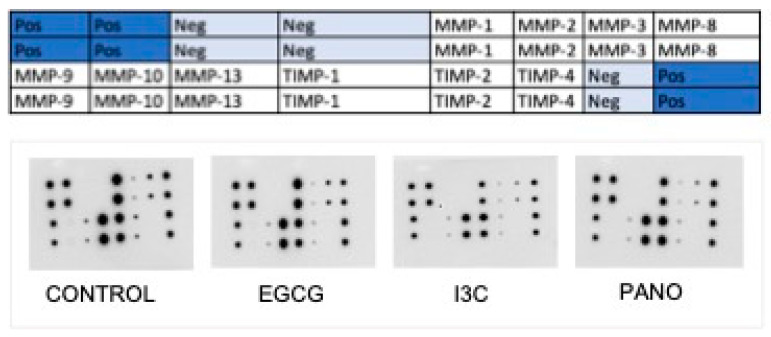
Protein array map of MMPs and TIMPs (**top row**) and experimental images (**bottom row**) of human specimen pretreated with epigenetic therapies. Conditioned medium collected from ovarian cancer human specimens pretreated with EGCG, I3C, and Pano was tested by protein array for cytokines and growth factors expression. See Materials and Methods.

**Table 1 biomolecules-13-01066-t001:** Analysis of cytokines and growth factors’ secretion following epigenetic treatments.

GF/CYTOKINE	FUNCTION	TREATMENTS
CONTROL	EGCG	I3C	PANO
MEAN, SD	MEAN, SD	EFFECT	MEAN, SD	EFFECT	MEAN, SD	EFFECT
IL-10	Induces cancer proliferation, migration, andimmunosuppression	183 ± 5	42 ± 4	77% 	50 ± 9.3	73% 	40 ± 1.5	78% 
IL-8	Stimulates ovarian cancer growth and proliferation	187± 0.25	150 ± 0.46	20% 	167 ± 5.7	11% 	139 ± 5	25% 
IL-11	Promotes chemoresistance, proliferation and survival of metastatic ovarian cancer cells	36 ± 3.6	18 ± 2.7	50% 	2.9 ± 2.4	92% 	0.8 ± 0.2	98% 
IL-6R	Correlated with aggressive behavior of high-grade ovarian cancer and poor prognosis when activated by IL-6	35.2 ± 3.6	41.6 ± 4.23	18% 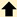	21.9 ± 2	38% 	34.78 ± 1.6	22% 
TNF-RI	Stimulates release of IL-6 and VEGF, induces peritoneal cancer spread	136 ± 2.2	30.5 ± 4.6	78% 	7.8 ± 2.5	94% 	12.2 ± 1.3	91% 

RANTES (CCL5)	Alters T-cells and NK cells function, induces cancer progression and resistance	136 ± 2.4	5.33 ± 1.3	96% 	38.8 ± 0.91	71% 	31.7 ± 1.46	77% 
MCP-1	Facilitates proliferation, migration, angiogenesis and immunosuppression	134 ± 2.3	135.5 ± 4.9	11% 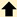	80.6 ± 4.3	40% 	81.7 ± 1.4	39% 

Optical Density (OD) signal was detected and compared between samples using Image J software. Mean and SD are represented in OD units, while effect is calculated in percent decrease (downward arrow) or increase (upward arrow) in relation to the control or drug free treatment group. Microsoft Word 16.73 was used to produce this table.

**Table 2 biomolecules-13-01066-t002:** Analysis of MMPs and TIMPs secretion following epigenetic treatments.

MMPs/TIMPs	FUNCTION	TREATMENTS
CONTROL	EGCG	I3C	PANO
MEAN, SD	MEAN, SD	EFFECT	MEAN, SD	EFFECT	MEAN, SD	EFFECT
MMP-1	Degradation of ECM	31,359 ± 518	33,999 ± 2477	8% 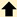	22,850 ± 1061	27% 	30,443 ± 2035	3% 
MMP-3	Remodeling of ECM	10,777 ± 752	10,209 ± 588	5% 	5391 ± 109	50% 	5083 ± 283	53% 
MMP-8	Stimulates inflammation, correlates with prognosis	23,777 ± 1229	16,732 ± 1456	30% 	19,900 ± 2347	16% 	25,445 ± 486	7% 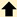
MMP-13	Stimulates metastasis, correlates with prognosis	4467 ± 331	3094 ± 411	31% 	2871 ± 910	36% 	2950 ± 186	34% 

TIMP-1	Inhibits metalloproteinase activity, cancer proliferation, migration, invasion	30,750 ± 1132	32,433 ± 1988	5% 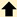	32,011 ± 2774	4% 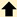	32,509 ± 988	6% 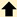
TIMP-2	Inhibits metalloproteinase activity, cancer proliferation, migration, invasion	23,552 ± 1099	25,728 ± 1834	9% 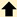	28,211 ± 4007	19% 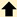	33,889 ± 636	43% 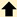

Optical Density (OD) signal was detected and compared between samples using Image J software. Mean and SD are represented in OD units, while effect is calculated in percent decrease (downward arrow) or increase (upward arrow) in relation to the control or drug free treatment group. Microsoft Word 16.73 was used to produce this table.

## Data Availability

Data is contained within the article and can be made available upon request.

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
