# Peer review of "The Effects of Natural Epigenetic Therapies in 3D Ovarian Cancer and Patient-Derived Tumor Explants: New Avenues in Regulating the Cancer Secretome"

_biomolecules, 2023, doi:10.3390/biom13071066_

Round 1

Reviewer 1 Report

This research study is very interesting and may help ovarian cancer patient treatment protocols improve and have better outcome. They have 18 Figures and two tables. Some of the figures can be used for supplementary figures, like fig 1. Although they have quantified data with statistical analysis, they did not use the actual data numbers, as notified within the review notes. They need to be included. Well reasoned, well laid out study, just couple typo highlighted in the revision.

Reviewer 3 Report

The manuscript by Kelly et al. aimed to test the anti-tumor effects of new generation epigenomic compounds in a 3D model of CAOV3 cells.

A lot of evidences showed that indeed these compounds have negative effects on the growth of these presented spheroids and their action is even better when used in combination with chemotherapy.

The validation of the data was done by using ovarian tumor explants alone. In addition, the secretome of the  tumor axplant s together with omental explants was somehow modulated by the epigenetic compounds maybe preliminary hypothesizing a mechanism of action of these epigenetic compounds since some cytokines such as IL-6 and IL-10, known to increase ovarian cancer cells growth, were found decreased upon epigenetic compound treatment alone, more if combined with chemotherapy. Accordingly to the changes in migration and invasion observed after treatment of ovarian carcinoma spheroids or explants, the levels of selected MMP or TIMP enzymes were found decreased.

The manuscript is interesting and represents a natural history for these groups after the paper also published on Biomolecules in 2021. Most of the experimental approaches were already described in the previous reports, however it seems that some data are nor really so quantitative unless the Authors explain better how they have, for example, evaluated ‘percent growth ’in Figs 1,3,8,9. Was this analysis always performed with MTT? On this regard, it could be useful repet the methodology in the figure legends also.

The dimension of the spheroids seem pretty big, I wonder whether these spheroid could be already apoptotic in the middle of the sphere. I say this because in both the pictures and the data reported in Fig 3 the conclusions seem not really quantitative. Could you make other analyses to morphologically characterize these spheroids like an confocal immunofluorescence? With this approach, the inner part of the spheroids could be better analyzed and maybe some apoptotic effect better analyzed.

Furthermore, comparing for example the picture of Figs 2 and 4, can you tell me if all the cells present around the spheres are apoptotic? Could be possible that cells just start to dissociate from the mainly object? If this is possible, I wonder whether you re-test these detached tumor cell populations after compound treatment.

Are you sure that you are testing organoids from the sample obtained from an ovarian carcinoma patient. Indeed, in the title ‘tumor explants’ are cited and not organoids.

The analysis of secretome showed interesting links between possible epigenetic modulations and soluble factors known to have an impact on ovarian cancer cell growth: maybe these experiments could be better introduced in Results section and in Discussion. Furthermore, secretome was analyzed only upon treatment of co-culterd omentum and tumor explants. Does this mean that tumor explants alone do not modulate the secretome upon treatment?

Minor comments:

Do yo need to add ethical issues for the use of a patient’s explants?

I found the introduction a little long while no word was spent in the first paragraph on PARP inhibitors while in lanes 101-104 you suppose the use of epigenetic compounds in BRCA positive tumors. Could you better introduce these concepts?

A lot of references are cited: I am not sure that they are all necessary. Please, check this point.

No word in Discussion section is mentioned about IL-11, while it seems the newest finding and REF #65 is not appropriate while another paper on Oncogene 2020 seems interesting and may help to make comments.

When more references are cited, maybe they can be order by publication date.

Round 2

Reviewer 2 Report

suitable for publication.

Author Response

Thank you for your response. Given that no further revisions were requested we do not have an attachment to provide. 

Reviewer 3 Report

The Authors seemed to regret to accomplish some of the issues made. Further explainations are below.

 2. The dimension of the spheroids seem pretty big, I wonder whether these spheroid could be already apoptotic in the middle of the sphere. I say this because in both the pictures and the data reported in Fig 3 the conclusions seem not quantitative. Could you make other analyses to morphologically characterize these spheroids like a confocal immunofluorescence? With this approach, the inner part of the spheroids could be better analysed and maybe some apoptotic effect better analysed.

 ‘however we chose to focus on morphological changes induced by our epigenetic treatments regarding apoptosis/necrosis for the purposes of this manuscript’.

This answer does not accomplish to the comments. The morphologic evaluation is not sufficient to demonstrate the efficacy of the relevant compounds.

 3. Furthermore, comparing for example the picture of Figs 2 and 4, can you tell me if all the cells present around the spheres are apoptotic? Could be possible that cells just start to dissociate from the mainly object? If this is possible, I wonder whether you re-test these detached tumor cell populations after compound treatment.

Yes, while part of the cells dissociating from the main 3D spheroid structures may be apoptotic, they can also be viable cells. We did observe de novo formation of new 3D spheroids from these viable detached cells under microscopy, corroborating the idea that viable cells are present and may represent micro disease in a clinical scenario.

If you have these data, it would be important to show it in this context.

7. I found the introduction a little long…….

I strongly suggest shrinking the Introduction section.

8. No word in Discussion section is mentioned about IL-11, while it seems the newest finding and REF #65 is not appropriate while another paper on Oncogene 2020 seems interesting and may help to make comments.

the answer to this comment is not sufficient. The decrease of the relevant cytokines would even increase the impact of the reported results but in lanes 1030-1039 only a repetition of the results is reported. I apologize since the article in Oncogene was published in 2020 Oncogene 2018 Jul;37(29):3981-3997. doi: 10.1038/s41388-018-0238-8. Epub 2018 Apr 17. Other information might be given about IL-6 and IL-10 in the context of aggressiveness, chemoresistance and immunorepression……..

None.
